# COMPOSING FEATURES: COMPOSITIONAL MODEL AUGMENTATION FOR STEERABILITY OF MUSIC TRANSFORMERS

## ABSTRACT

Music is a combinatorial art. Given a starting sequence, many continuations are possible, yet often only one is written down. With generative models, we can explore many. However, finding a continuation with specific combinations of features (such as rising pitches, with block chords played in syncopated rhythm) can take many trials. To tackle the combinatorial nature of composing features, we propose a compositional approach to steering music transformers, building on lightweight fine-tuning methods such as prefix tuning and bias tuning. We introduce a novel contrastive loss function that enables us to steer compositional models over logical features using supervised learning. We examine the difficulty in steering based on whether features musically follow a prime or not, using existing music as a proxy. We show that with a relatively small number of extra parameters, our method allows bias tuning to perform successful fine-tuning in both the single-feature and compositional setting.

## 1 INTRODUCTION

Recent research has focused on ways of adapting unconditional language models to perform well on conditional tasks they were not initially trained on. Methods such as fine-tuning, side-tuning, bias-tuning, and prompt or prefix tuning have emerged as lead candidates for such tasks such as steering the sentiment of or words mentioned in a sentence (Ben-Zaken et al., 2021; Li & Liang, 2021; Lester et al., 2021; Dathathri et al., 2020). However, such tasks are normally presented in a non- or minimally-compositional approach (possibly controlling for two orthogonal variables, such as sentiment and topic (Keskar et al., 2019), but rarely more than that). In contrast, in the domain of music generation, sequence-continuation is inherently a highly compositional problem: the user likely has many aspects of the output they would like to control, such as speed, dynamics, harmony, or texture, each of which can be decomposed into multiple sub-features. Furthermore, the relationship between these features and the output is more diffuse than in the NLP settings: while in examples such as Keskar et al. (2019), individual words indicate the different conditions being satisfied, in music the entire sequence of tokens are used in evaluating a single feature (for instance, average pitch over a span of time depends on all tokens representing that span). This makes compositionality even more challenging than in the text generation setting. On the other hand, music is a sequential domain where there are clear logical features which can be examined, such as average dynamics or number of notes per second. This further motivates using music as a test-bed for highly compositional tasks involving simultaneously steering an autoregressive model towards several particular desired attributes according to different classifiers.

As a more immediate motivation, consider the following scenario: a composer wants to sample from the pre-trained Music Transformer (Huang et al., 2018) to complete a musical phrase. In addition to the overall musical quality of the continuation, they want it to stay in key *and* switch to using block chords (i.e. a few notes played together at once). Repeatedly sampling continuations from the model and cherry-picking a good sample (*rejection sampling*) would be very labor-intensive, but assuming they can compute binary features for "stays in key" and "uses block chords", the composer could sample a large number of continuations and cherry-pick from the smaller set of continuations which exhibit all features (thereby delegating part of the accept/reject step). While significantly less labor-intensive, this solution could potentially require sampling enormous amounts of continuations

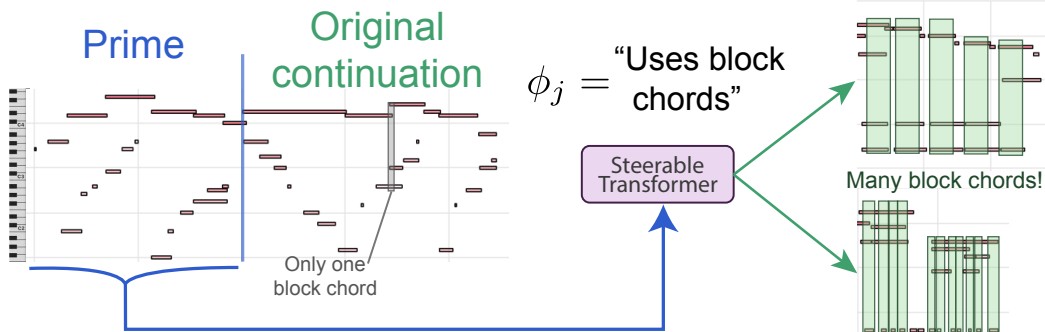

Figure 1: Showcasing a prime from the test set steered in two different ways. The original continuation has arpeggiation (the steep rising and falling lines) throughout. The right piano rolls were steered with a logical feature that checks for many simultaneous notes (block chords). In particular, the bottom right piano roll shows that the model learned to "exploit" the logical feature by repeating a low note, while still sounding musical.

if, for instance, [6] the requested features significantly differ from those of the priming sequence (such that it and the desired continuation form an unlikely sequence according to the generative model). In other words, the pre-trained model could be a poor proposal distribution for some applications.

In this work, we are interested in developing better proposal distributions through an adaptation approach which can steer the generative model towards continuations which i) are significantly more likely to exhibit the requested feature, and ii) exhibit a satisfactory musical quality. The approach should be able to accommodate a large number of features without adding significant memory or computation overhead. We achieve this by making features composable, making it possible to steer features independently and also multiple features at once. Figure 1 shows that when using a pretrained transformer model augmented with a relatively small number of additional parameters, we are able to steer towards arbitrary logical music features and achieve realistic music generation simply by sampling directly from the model [1]. In contrast, the same unconditional transformer model fails to produce any examples satisfying those features even when generating 100 samples. With our method, a composer can control both the short-chunk features, and the long-form structure, by chaining together chunks steered in different directions (using different combinations of features), while maintaining long-term coherence (by leveraging the transformers' full self-attention receptive field). See the demo in the footnote for longer steered examples, and compositions semi-automatically generated using our feature tuning approach in a musician-directed way.

## 2   PROBLEM FORMULATION

Music Transformer is an autoregressive language model which decomposes the joint probability of a sequence of tokens $x_1, \ldots, x_N$ (where $x_n \in \mathcal{K}$, and $\mathcal{K}$ is a set of categorical tokens) into

$$p(\mathbf{x} = x_1, \ldots, x_N) = p(x_1) \prod_{n=2}^{N} p(x_n \mid x_1, \ldots, x_{n-1}). \tag{1}$$

It leverages a common modeling approach which represents the conditional probabilities $p(x_n \mid x_1, \ldots, x_{n-1})$ using a neural network (Bengio et al., 2003). As its name implies, Music Transformer uses a Transformer network architecture (Vaswani et al., 2017a). Each token $x_n$ is first mapped to a real-valued embedding $\mathbf{e}_n$ (for instance using a lookup table), then the network maps each sequence $\mathbf{e}_1, \ldots, \mathbf{e}_{n-1}$ to a probability distribution for the value of $x_n$ over the elements of $\mathcal{K}$.

Sequence continuation in an autoregressive language model works by repeatedly sampling from its distribution over the next token given the previous tokens. Starting from some priming se-

---

[1]Listen to these examples at `https://storage.googleapis.com/composing-features/index.html`

quence $\mathbf{x}_p = (x_0, \ldots, x_M)$, we first sample $y_{M+1} \sim p(\cdot \mid x_0, \ldots, x_M)$, then $y_{M+2} \sim p(\cdot \mid x_0, \ldots, x_M, y_{M+1})$, and so on, until the end of the continued sequence $\mathbf{x}_c = (y_{M+1}, \ldots, y_N)$. [2] Many downstream tasks can be cast as sequence continuation problems, including the steerable music generation problem investigated in this work.

We are given a set of features $\Phi = \{\phi_j\}_{j=1}^J, \phi_j \in \mathcal{K}^N \to \{0, 1\}$. Each $\phi_j$ takes the value 1 if a prime-continuation pair exhibits that feature (which we note as $(\mathbf{x}_p, \mathbf{x}_c) \models \phi_j$), and 0 otherwise. Note that the features must take both prime and continuation sequences as input, since some continuation features may be relative to the priming sequence (e.g. "significantly higher pitch"). Our true objective with respect to feature $\phi_j$ is to steer the model towards a distribution which maximizes

$$\mathbb{E}_{\mathbf{x}_c \mid \mathbf{x}_p} \big[ (\mathbf{x}_p, \mathbf{x}_c) \models \phi_j \big] \tag{2}$$

while maintaining musicality. This objective is non-differentiable because $(\mathbf{x}_p, \mathbf{x}_c) \models \phi_j$ is a non-differentiable satisfiability criterion.

In addition to the single-feature problem, we also consider the problem of *composed* features $\hat{\Phi}$, i.e.

$$(\mathbf{x}_p, \mathbf{x}_c) \models \hat{\Phi} \quad \equiv \quad \bigwedge_{\phi_j \in \hat{\Phi} \subseteq \Phi} (\mathbf{x}_p, \mathbf{x}_c) \models \phi_j, \tag{3}$$

to account for scenarios where a user is interested in steering the model towards multiple features (such as in the "stays in key" and "exhibits extreme dynamical contrast" scenario dicussed in the introduction).

## 3 PROPOSED APPROACH

We start by describing our proposed approach in the single-feature case and later on explain how we adapt it to the compositional case.

### 3.1 LIKELIHOOD-BASED TRAINING

While approaches using reinforcement learning—such as KL-regularized deep Q-learning (Jaques et al., 2016)—could be used to overcome the non-differentiability problem, in this work we consider a proxy loss in the form of the negative log-likelihood

$$l = -\log p_\theta(\mathbf{x}_c \mid \mathbf{x}_p), \tag{4}$$

which we use in two ways:

1. **Positively**: given a prime–continuation pair $(\mathbf{x}_p, \mathbf{x}_c) \models \phi_j$, we find an adaptation $\theta_j$ of the model's parameters $\theta$ that minimizes $l_{\checkmark}$ (we use the symbol $\checkmark$ to denote the fact that $l$ is computed using the correct parameters $\theta_j$). By using prime–continuation examples that sound musical, we ensure that the steered model stays musically grounded.

2. **Negatively**: we can also take advantage of other features $\phi_i$ for which $(\mathbf{x}_p, \mathbf{x}_c) \not\models \phi_i$, by maximizing $l_{\times}$ (we use the symbol $\times$ to denote the fact that $l$ is computed using the incorrect parameters $\theta_i$).

The positive case corresponds to maximum-likelihood training. Additionally, we can exploit the intuition that the adapted parameters $\theta_j$ should "explain" the prime–continuation pair $(\mathbf{x}_p, \mathbf{x}_c) \models \phi_j$ better than $\theta_i$ (for some feature $\phi_i$ for which $(\mathbf{x}_p, \mathbf{x}_c) \not\models \phi_i$) or $\theta$ (the non-adapted model parameters, with a corresponding loss $l_\emptyset$). In other words, we can maximize the probability of choosing $\theta_j$ over $\theta_i$ and $\theta$ by minimizing a contrastive loss of the form

---

[2]To simplify the discussion, we assume a fixed sequence length $N$, but the explanation applies to sequences of varying lengths as well.

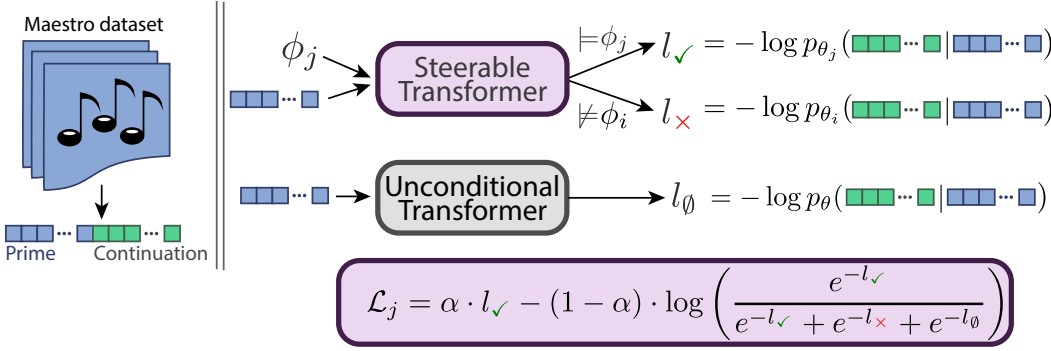

Figure 2: Overall training setup, where the (positive) feature being learned is $\phi_j$ (with parameters $\theta_j$), and the negative feature is $\phi_i$ (with parameters $\theta_i$). The parameters $\theta$, used for the unconditional negative log-likelihood, are not learned.

$$-\log\left(\frac{p_{\theta_j}(\mathbf{x}_c \mid \mathbf{x}_p)}{p_{\theta_j}(\mathbf{x}_c \mid \mathbf{x}_p) + p_{\theta_i}(\mathbf{x}_c \mid \mathbf{x}_p) + p_\theta(\mathbf{x}_c \mid \mathbf{x}_p)}\right) = -\log\left(\frac{e^{-l_\checkmark}}{e^{-l_\checkmark} + e^{-l_\times} + e^{-l_\emptyset}}\right) \quad (5)$$

We propose a loss that interpolates between Equations 4 and 5 using an $\alpha$ coefficient (which is treated as a hyperparameter):

$$\mathcal{L}_j = \alpha \cdot l_\checkmark - (1 - \alpha) \cdot \log\left(\frac{e^{-l_\checkmark}}{e^{-l_\checkmark} + e^{-l_\times} + e^{-l_\emptyset}}\right) \quad (6)$$

Intuitively, the maximum-likelihood setting should suffice to achieve our adaptation goals, but in practice we find that the approach benefits from the inclusion of negative cases through a contrastive loss term. We tried different $\alpha$ values and found $\alpha = 0.8$ to work well in practice. See Figure 2 for an illustration of the training setup.

## 3.2 FEATURE-CONDITIONAL ADAPTATION

Fine-tuning all model parameters can be prohibitive if the number $J$ of features is large (let alone combinatorially large in the compositional case); however, recent work provides effective and lightweight alternatives:

1. **Prefix tuning** (Li & Liang, 2021) works by preprending learnable task embeddings $\mathbf{e}_{-K}, \ldots, \mathbf{e}_{-1}$ to the priming sequence embeddings $\mathbf{e}_1, \ldots, \mathbf{e}_M$. The loss gradient is then backpropagated through the language model and into the task embeddings.

2. **Bias-tuning** (Ben-Zaken et al., 2021) works by adapting a small subset $\theta_b \subset \theta$ of the transformer's parameters, namely the biases of its affine transformations. Since these biases amount to a small fraction of the model's parameters, in the case where the number of tasks is relatively small, tuning separate $\theta_b$ for each task becomes feasible. We present an extension to bias-tuning where the number of tasks is exponential in the number of total classification functions, using an approach which nevertheless only requires a number of tuned parameters linear in the number of total classification functions.

In practice, while prefix tuning showed promise in the single-feature setting, we were unable to make it work in the compositional setting. We therefore focus our investigation on bias-tuning and present prefix tuning results in the Appendix.

In the compositional setting, a naive approach requires learning $2^{|\Phi|} - 1$ model adaptations. Instead, we propose to express the adaptation for a composed feature $\hat{\Phi}$ as the combination of the $\theta_j$ of its underlying features $\phi_j \in \hat{\Phi}$. More specifically, for bias-tuning we average the adapted biases as

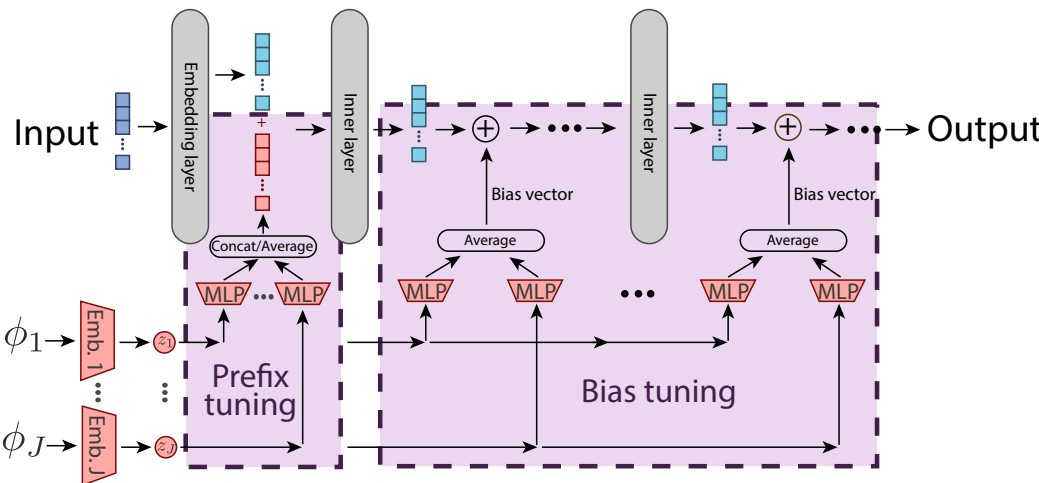

Figure 3: Prefix- and bias-tuning methods used to steer the music transformer.

$$\theta_b = \frac{1}{|\hat{\Phi}|} \sum_{\phi_j \in \hat{\Phi}} \theta_{b,j}. \tag{7}$$

Note that we do not simply use the above heuristic to compose feature adaptations *post-hoc*; we *train* the model in a compositional setting (by sampling prime–continuation pairs $(\mathbf{x}_p, \mathbf{x}_c) \models \hat{\Phi}$ for various composed features) so that the single-feature adaptations can learn to work well in conjunction with each other. See Figure 3 for an illustration of the methods.

## 4 METHODOLOGY

### 4.1 MUSICAL FEATURES

We consider 18 features representing properties that a composer would typically like to control; see the Appendix for a complete list. Features include both *absolute* and *relative* features. Absolute features apply only to the continuation, while relative features, such as "the continuation is significantly higher in pitch than the prime" or "the continuation is significantly more rhythmically dense than the prime" describe relations between the prime and continuation. Note that this type of model should work with any feature function which takes in sequences and returns a boolean; we chose examples that have clear musical meanings and are easy to implement.

In addition to exploring binary musical features, we also explore fine-tuning to a provided dataset; in our experiments we fine-tune to the Maestro dataset (Hawthorne et al., 2019), and refer to this feature as $\phi_{\maestro}$.

### 4.2 TRAINING DATA

The prime–continuation pairs used to train the feature-specific adaptations are drawn from the Maestro dataset (Hawthorne et al., 2019), an open-source collection of virtuoso piano performances. Each prime is 100 tokens long (approximately 2-10 seconds), and each continuation is also 100 tokens long. Note that since all prime–continuation pairs are drawn from the Maestro dataset, we implicitly assume that $(\mathbf{x}_p, \mathbf{x}_c) \models \phi_{\maestro}$ always holds.

In the single-prefix setting, when a prime–continuation pair $(\mathbf{x}_p, \mathbf{x}_c)$ is sampled for feature $\phi_j$, we also select another feature $\phi_i$ at random such that $(\mathbf{x}_p, \mathbf{x}_c) \not\models \phi_i$ and use its corresponding $\theta_i$ in Equation 6.

In the compositional case, we first extend $\Phi$ so that each feature has a corresponding "negated" feature (e.g. "has both loud and soft pitches" vs "no contrast in dynamics"), bringing the total

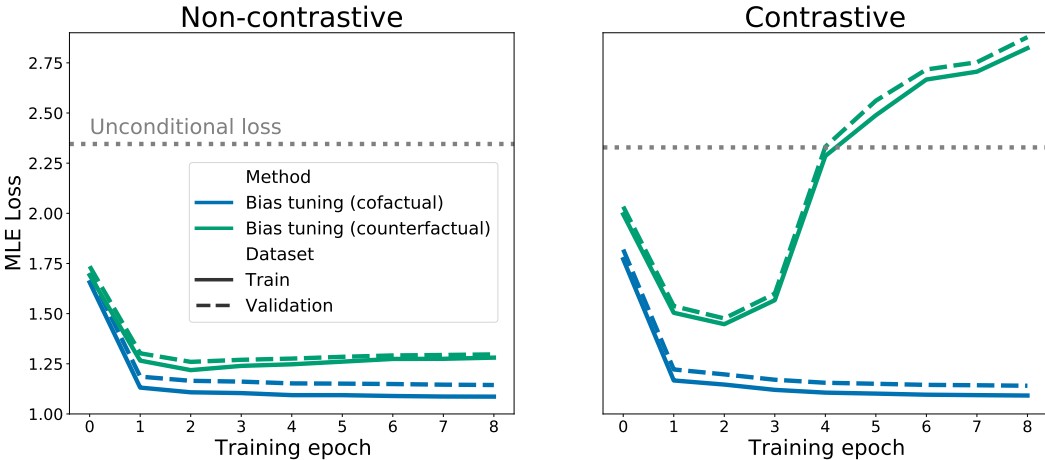

Figure 4: MLE loss during training of the bias tuning method. Using the contrastive loss is more effective at discriminating between cofactual and counterfactual features (see subsection 5.2).

number of features to $2 \cdot |\Phi| = 36$. By definition, each prime–continuation pair exhibits exactly 18 features. When sampling a pair $(\mathbf{x}_p, \mathbf{x}_c)$, we compute $l_\checkmark$ with respect to a random subset of those 18 features (with size drawn at random from $U[1, 12]$). We compute $l_\times$ by negating some of the sampled features.

### 4.3 IMPLEMENTATION

In the single-feature case, we use a standard prefix tuning model as per Li & Liang (2021), associating every feature $\phi_j$ with a prefix $v_j$. As in their paper, we used the finding that a relatively low-dimensional embedding space of feature prefixes (200 dimensions) projected to a higher-dimensional embedding (2048 dimensions) using a shared MLP across feature prefixes worked well. The pre-trained transformer model had no parameters modified, but was changed to accept these 2048-dimensional vectors as 4 512-dimensional vectors prepended before the 512-dimensional vectors representing the input tokens (as the transformer's latent embedding was 512 dimensions). The resulting sequence of vectors were masked using causal attention in order to predict $\mathbf{x}_c$.

The loss used in the compositional setting is almost identical to Equation 6, the main difference being in the contrastive loss implementation. The features used to compute $l_\times$ are obtained by negating some of the features used to compute $l_\checkmark$, and the fraction $\beta$ of negated features is used to modulate the interpolation coefficient as $\alpha' = \alpha - \min(\alpha, \beta)$. The intuition is that the differences in steering should be more noticeable when comparing feature sets with less overlap.

## 5 RESULTS—SINGLE-FEATURE REGIME

### 5.1 TRAINING STATISTICS

As seen in Figure 4, in the contrastive case the model not only learns to assign progressively more weight to $(\mathbf{x}_p, \mathbf{x}_c | v_j)$ (the "positive" examples), but also learns to assign progressively less weight to $(\mathbf{x}_p, \mathbf{x}_c | v_i)$ (the "negative" examples).

### 5.2 THE LARGE VARIANCE IN DIFFICULTY OF PRIME-FEATURE PAIRS

As shown in Figure 6, there is a large difference in the efficacy of rejection sampling with respect to primes from the Maestro validation dataset which are followed in the Maestro validation dataset by a satisfying continuation (here termed "cofactual primes"), and rejection sampling with respect to primes from the maestro validation dataset which in fact are not followed by a satisfying continuation ("counterfactual primes"). In fact, for some continuations the "counterfactual" probability of

|                | Prefix tuning (Contrastive) | Prefix tuning (Non-contrastive) | Unconditional |
|----------------|------------------------------|----------------------------------|---------------|
| Cofactual      | **0.58 (0.57, 0.59)**        | 0.54 (0.53, 0.55)                | 0.38 (0.37, 0.39) |
| Counterfactual | **0.39 (0.38, 0.40)**        | 0.33 (0.32, 0.34)                | 0.23 (0.22, 0.24) |
| Random         | **0.43 (0.42, 0.44)**        | 0.40 (0.39, 0.41)                | 0.27 (0.26, 0.27) |

Table 1: Comparison of the efficacy of Prefix tuning (using contrastive and non-contrastive) with the efficacy of the unconditional approach. The numbers in parentheses indicate 95% confidence intervals.

efficacy is almost 0%. Prefix tuning not only increases the efficacy of steering overall, but also tends to decrease the relative difficulty of "counter-factual" steering.

### 5.3 CONTRASTIVE LOSS GENERALLY IMPROVES EFFICACY OVER NON-CONTRASTIVE LOSS

While efficacy differs for different primes and features, contrastive learning generally significantly outperforms non-contrastive learning, which almost universally outperforms the unconditional model. As seen in Table 1, it is usually (but not always) the case that the efficacy of the model trained with contrastive loss is significantly higher than that trained with non-contrastive loss, and very rarely do both contrastive and non-contrastive models fail (i.e., fail to perform significantly better, and actually seem to do worse, than the unconditional model).

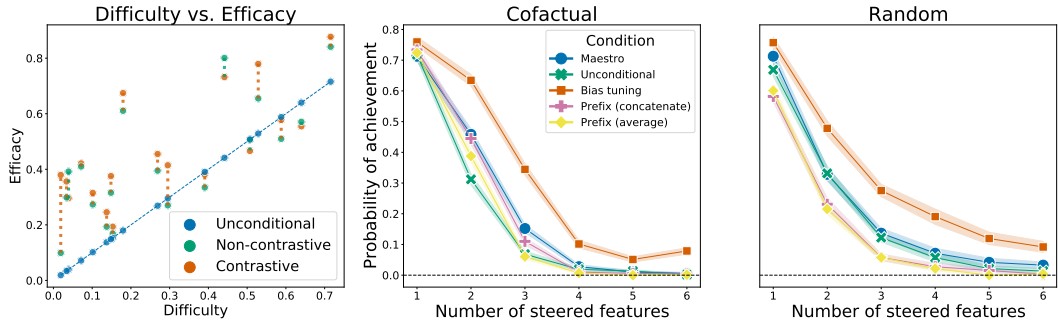

Figure 5: **Left:** Difficulty is quantified as the efficacy of the unconditional model. In most cases, contrastive learning significantly improves over non-contrastive learning (as evidenced by the dashed orange lines). In only a few cases does prefix tuning fail to improve over the unconditional model. **Middle and right:** Efficacy of various methods with various numbers of steered features on cofactual and random features.

## 6 RESULTS IN THE COMPOSITIONAL SETTING

**Steering unconditionally becomes intractable as the number of steered features increases.** Under a naive assumption of independence among feature presence, one would assume that the rejection sampling efficacy in steering a total of $n$ features would roughly equal $p^n$, where $p$ is the mean probability of a feature being satisfied. While this assumption is naive, we do find that in the unconditional setting, the level of efficacy of unconditional sampling drops substantially as the number of features to be steered increases. On average, only 0.2% of the time were 6 *cofactually steered* features all achieved correctly using rejection sampling.

**Trained models degrade at a much slower rate as the number of steered features increases.** While the bias-tuning model had average efficacy of 59% in the single prefix setting with randomly sampled prefixes, it dropped by an order of magnitude (to 7.9%) in the randomly-sampled 6-prefix setting (still significantly outperforming unconditional rejection sampling).

**Bias-tuning is the most effective way to steer a model as the number of steered features increases.** This was true both for "cofactual" steered features and random steered features (although

random steered features understandably was a harder task overall). Notably, even though prefix tuning had performed well when trained on single prefixes, it failed to generalize to the compositionally-trained case *even in the cases that a single prefix was being steered.*

**Listening test**  The above experiments evaluate the effectiveness of methods in satisfying the requested features. To evaluate the musical quality of these 'successful' generations, we conducted a listening study with musicians to compare the steering of three features between our overall best augmentation approach (i.e. bias tuning) and the baseline of rejection sampling on the unconditional model. Questions were posed as pairwise comparisons, and listeners were asked to rate which one they thought was more musical. To prepare the samples, we randomly picked 120 primes from the test set. For steering each prime, we randomly chose three-feature sets that were present in the dataset as the conditioning features. We asked eight musicians to each rate fifteen pairs. The results show that our bias-tuning approach was much preferred over the unconditional, and the results were statistically significant ($p < 0.0003$). Bias tuning won 63 of the pairwise comparisons, tied for 26 pairs, and lost for 31 pairs. This shows that our bias-tuning approach is not only more effective in steering features, but also produces musically more compelling results.

## 7 FUTURE WORK

As prior work demonstrates, there are several possible avenues for steering transformers. We have established a baseline for compositional steering with bias-tuning and shown negative results on compositional steering with prefix tuning, but believe novel loss functions, architectures, and decoding strategies could improve on these methods. In addition, other methods for steering, such as Plug-and-Play, side-tuning or CTRL can be applied to this area with potentially promising results.

This study focused on binary features, but most of those binary features were actually derived from cutoffs on continuous features, like amount of dynamic contrast or percentage pitch increase. Future work could include features which are associated with a continuous value from 0 to 1.

## 8 RELATED LITERATURE

This work would be impossible without the wealth of research on transformer models for sequence generation tasks. Starting with Vaswani et al. (2017a), researchers realized that this paradigm enables far more coherent and diverse generation than the recurrent neural networks typically used before (Sherstinsky, 2018). Another milestone was the development of GPT-3, which showed that with sufficient size/training data such models could potentially perform few shot learning (Brown et al., 2020). Several subsequent papers, however, demonstrated the inadequacy of few-shot learning for many tasks (Perez et al., 2021). A few alternatives to online few-shot learning have since emerged.

Prompt tuning was initially explored ad-hoc in the context of finding ways to produce interesting output by GPT-3, and was formalized by Lester et al. (2021). While originally prompts were designed as tokens from the transformer's vocabulary, subsequent studies generalized them to any embeddings prepended to the input (Li & Liang, 2021). We extend research into prefix tuning by considering aggregation methods among compositional prefixes.

Feature-wise transformations, i.e. elementwise scaling and/or biasing of features in a network based on side-information, have been applied in a wide variety of problem settings—see Dumoulin et al. (2018) for an overview. We draw direct inspiration from BitFit's bias-tuning approach (Ben-Zaken et al., 2021) and cast it as a feature-wise transformation approach. By factoring the additive perturbations in Figure 3 into their preceding layers, our bias-tuning implementation can be described as a multi-task variant of BitFit where the parameterizations are tied across features. A key difference is that our feature-specific adaptations are designed to be composable, which to our best knowledge has not yet been explored in the context of large language models—although compositional adaptations using feature-wise multiplicative interactions have been studied in the context of zero-shot image classification (Purushwalkam et al., 2019). Similarly, side-tuning, or summing task-specific features with general language-model features, has shown significant enhancements in few-shot learning, but is typically not performed compositionally (Zhang et al., 2019).

Contrastive learning is used in representation learning to train a network which maps "similar" (positive) inputs to nearby representations and "dissimilar" (negative) inputs far away from the positive inputs. See Le-Khac et al. (2020) for a theoretical framework and overview. In generative modeling, contrastive divergence (Hinton, 2002) was proposed to train Restricted Boltzmann Machines (Smolensky, 1986), image-to-image translation models (Baek et al., 2021; Park et al., 2020; Liu et al., 2021), and conditional (Kang & Park, 2020) and unconditional (Jolicoeur-Martineau, 2018) generative adverarial networks (Goodfellow et al., 2014). Our contrastive formulation differs from previous work in two ways. First, rather than selecting positive and negative examples related to the conditioning signal (musical features in our case) and using the contrastive loss to predict which example "agrees" with the signal, we select positive and negative *conditioning signals* (i.e. different musical features) and use the contrastive loss to predict which conditioning signal explains the prime–continuation pair best. Second, we also treat the absence of a conditioning signal (i.e. the original generative model) as a negative conditioning signal, meaning that we want the model conditioned on the "correct" musical features to explain the prime–continuation pair better than the unconditional model.

Other work leveraging language models for multiple tasks include CTRL (Keskar et al., 2019) and Plug-and-Play models (Dathathri et al., 2020). However, CTRL has the disadvantage that it requires knowing the tasks during the training of the large language model, while Plug-and-Play requires multiple passes through the language model, which can be expensive for sufficiently large models.

**Background in controllable deep generative models for music** Advances in sequence modeling (van den Oord et al., 2016; Vaswani et al., 2017b; van den Oord et al., 2017; Choromanski et al., 2020) has enabled long-form music generation in both the symbolic domain (Oore et al., 2020; Huang et al., 2019; Payne, 2019; Liutkus et al., 2021; Hsiao et al., 2021) and the audio domain (van den Oord et al., 2016; Hawthorne et al., 2019; Dieleman et al.; Dhariwal et al., 2020).

Similar to language, researchers in music generation have been adapting these language models towards controllable generation, such as by conditioning on one part of a musical piece to complete the rest, such as melody harmonization (Simon et al., 2008; Liang, 2016; Choi et al., 2020), or more generally arbitrary partial score completion (Huang et al., 2017; Hadjeres et al., 2017). Representational learning approaches such as autoencoders (AEs) and variational autoencoders (VAEs) have also been used for steering interpolations or transformations along learned latent dimensions, a low-level disentangled attribute-based dimension such as note density (Roberts et al., 2018; Kawai et al., 2020), or a high-level learned dimension such as energy level in mood that is then realized through its mapping to low-level features such as note-density and rhythm (Tan & Herremans, 2020), controlling chord progressions and texture independently (Simon et al., 2018; Wang et al., 2020), or rearranging a piece to have increased polyphony or rhythmic density (Wu & Yang, 2021).

Controllable generative models typically does not offer users full control (i.e. only allows users to specify a small number of low-level or high-level controls, or through an example), while relying on its learned stylistic distribution and/or features encoded from the user-specified template piece to fill in the rest of the musical details. In contrast, traditional constraint satisfaction based music generation systems do not have prior knowledge of the desired stylistic distribution, instead rely on users to specify a large number of musical constraints to guide its search (see Pachet & Roy (2001) for a survey). When using the former systems, users may still feel a lack of agency, while the latter can impose a laborious process. Our approach explores the space in between, allowing users to compose multiple features along different musical dimensions for short chunks (similar to constraint specification), while leveraging pretrained transformers' expressiveness to aid users in maintaining coherence in virtuosic long-form composition.

## 9 CONCLUSION

We have shown that music transformers can be directed towards a specific generative "task" using some of the same methods as natural language transformers. In addition, we have studied compositionality in this domain. Compositionality (including relatively high levels of compositionality) is critical in the music domain (and other domains) if the user wants control over the output. We establish that compositionality is a hard problem, and propose adaptations of several solutions from the literature (bias-tuning and prefix-tuning) to address these challenges. We find success with bias-tuning, but not with prefix-tuning. While our results are promising, there is clearly significant room for improving on the efficacy of steering a transformer compositionally.

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

## A MUSICAL FEATURES

Note that while some features may appear to be opposites (e.g., "loud" vs "soft"), and while it is true that they are mutually exclusive, in fact it is possible for a sequence to satisfy neither (e.g., if it's in a middle dynamic level).

Absolute features:

1. "Loud" - minimum velocity is greater than 60
2. "Soft" - maximum velocity is less than 60

3. "Has Dynamic Contrast" - Extreme Dynamic Contrast" - The sequence has two notes whose velocities differ by more than 30

4. "Extreme Dynamic Contrast" - The sequence has two notes whose velocities differ by more than 70

5. "All consonances" - the sequence has no dissonances (simultaneous notes with an absolute difference modulo 12 of 1, 2, 10, 11, or 6)

6. "Long sharp dissonance" - the sequence has a sharp dissonance (simultaneous notes being played with a difference of 1 or 11) that lasts for a significant amount of time

7. "Only melody" - only a single note is playing at a time

8. "Few onsets" - when grouping notes according to attack time, there are few groups per second

9. "Many onsets" - when grouping notes according to attack time, there are many groups per second

10. "Blocks of two" - there are groups of two notes being played simultaneously

11. "Larger blocks" - there are blocks of three or more notes being played simultaneously

12. "Within single key" - all notes fit within a single major scale

Relative features:

1. "Significantly lower average pitch than prime"

2. "Significantly higher average pitch than prime"

3. "Significantly higher number of grouped attacks than prime"

4. "Significantly lower number of grouped attacks than prime"

5. "Significantly more notes per second than prime"

6. "Significantly fewer notes per second than prime"

7. "Could fit in the same key as the prime"

8. "Has 2 or more pitch classes (pitch mod 12) which the prime doesn't have"

## B    ADDITIONAL FIGURES

## C    ON A PER-FEATURE LEVEL, THERE IS A BENEFIT TO INCLUDING AN ADDITIONAL STEERED FEATURE IN CONJUNCTION WITH $n$ OTHER STEERED FEATURES AS OPPOSED TO REJECTION SAMPLING $+ n - 1$ STEERED FEATURES UP TO $n = 6$.

As seen in Figure 8, the benefit of steering doesn't only come from an increased likelihood of a single feature (which in fact would make the joint likelihood of 6 total features higher, but would not indicate increased compositionality). Rather, for a single constant feature, bias-tuning towards that feature increases steerability even in the presence of 5 other steered features.

| Feature | | Prefix tuning (Contrastive) | Prefix tuning (Non-contrastive) | Unconditional |
|---|---|---|---|---|
| all_consonances | Cofactual | **0.51 (0.48, 0.54)** | 0.31 (0.27, 0.35) | 0.37 (0.34, 0.40) |
| | Counterfactual | **0.37 (0.34, 0.40)** | 0.23 (0.19, 0.27) | 0.28 (0.25, 0.31) |
| | Random | 0.30 (0.26, 0.34) | 0.27 (0.23, 0.31) | 0.17 (0.14, 0.20) |
| contrasting_dynamics | Cofactual | 0.78 (0.75, 0.82) | 0.82 (0.79, 0.86) | 0.52 (0.48, 0.56) |
| | Counterfactual | 0.68 (0.64, 0.72) | 0.74 (0.71, 0.78) | 0.37 (0.33, 0.41) |
| | Random | 0.73 (0.69, 0.77) | **0.83 (0.80, 0.87)** | 0.44 (0.39, 0.48) |
| extreme_dynamic_contrast | Cofactual | 0.36 (0.32, 0.40) | **0.45 (0.40, 0.49)** | 0.08 (0.06, 0.10) |
| | Counterfactual | 0.28 (0.24, 0.32) | 0.33 (0.29, 0.37) | 0.02 (0.01, 0.04) |
| | Random | 0.25 (0.21, 0.28) | **0.39 (0.35, 0.44)** | 0.01 (0.00, 0.02) |
| few_onsets | Cofactual | **0.58 (0.54, 0.63)** | 0.49 (0.45, 0.54) | 0.06 (0.04, 0.09) |
| | Counterfactual | 0.21 (0.18, 0.25) | 0.16 (0.12, 0.19) | 0.02 (0.00, 0.03) |
| | Random | 0.27 (0.23, 0.31) | 0.25 (0.21, 0.29) | 0.02 (0.01, 0.03) |
| long_sharp_dissonance | Cofactual | **0.36 (0.32, 0.40)** | 0.14 (0.11, 0.17) | 0.04 (0.02, 0.05) |
| | Counterfactual | **0.38 (0.34, 0.43)** | 0.06 (0.04, 0.08) | 0.01 (0.00, 0.02) |
| | Random | **0.40 (0.35, 0.44)** | 0.10 (0.07, 0.12) | 0.01 (-0.00, 0.01) |
| loud | Cofactual | 0.87 (0.84, 0.89) | 0.84 (0.81, 0.88) | 0.79 (0.75, 0.82) |
| | Counterfactual | **0.73 (0.69, 0.77)** | 0.55 (0.51, 0.59) | 0.35 (0.31, 0.39) |
| | Random | **0.74 (0.70, 0.78)** | 0.57 (0.53, 0.61) | 0.44 (0.40, 0.49) |
| many_verticalities | Cofactual | 0.82 (0.79, 0.86) | 0.78 (0.75, 0.82) | 0.21 (0.18, 0.25) |
| | Counterfactual | 0.50 (0.45, 0.54) | 0.44 (0.39, 0.48) | 0.13 (0.10, 0.16) |
| | Random | **0.70 (0.66, 0.74)** | 0.61 (0.57, 0.65) | 0.20 (0.16, 0.23) |
| potentially_same_key | Cofactual | 0.64 (0.60, 0.68) | 0.55 (0.51, 0.59) | 0.65 (0.61, 0.69) |
| | Counterfactual | 0.18 (0.15, 0.21) | 0.17 (0.13, 0.20) | 0.18 (0.15, 0.22) |
| | Random | 0.32 (0.28, 0.36) | 0.29 (0.25, 0.33) | 0.34 (0.30, 0.38) |
| relative_pc_shift | Cofactual | 0.52 (0.48, 0.56) | 0.47 (0.43, 0.52) | 0.34 (0.30, 0.38) |
| | Counterfactual | **0.41 (0.37, 0.45)** | 0.31 (0.27, 0.35) | 0.20 (0.16, 0.23) |
| | Random | 0.44 (0.39, 0.48) | 0.40 (0.36, 0.44) | 0.26 (0.23, 0.30) |
| significantly_higher_center_of_mass | Cofactual | 0.26 (0.22, 0.29) | 0.23 (0.19, 0.26) | 0.21 (0.18, 0.25) |
| | Counterfactual | 0.17 (0.14, 0.20) | 0.13 (0.11, 0.16) | 0.10 (0.08, 0.13) |
| | Random | 0.15 (0.12, 0.18) | 0.14 (0.11, 0.17) | 0.14 (0.11, 0.17) |
| significantly_higher_note_density | Cofactual | 0.64 (0.60, 0.68) | 0.64 (0.60, 0.69) | **0.72 (0.68, 0.76)** |
| | Counterfactual | 0.50 (0.46, 0.55) | 0.50 (0.46, 0.55) | **0.61 (0.57, 0.65)** |
| | Random | 0.52 (0.48, 0.56) | 0.57 (0.52, 0.61) | 0.59 (0.55, 0.64) |
| significantly_higher_rhythmic_density | Cofactual | 0.58 (0.54, 0.62) | 0.59 (0.55, 0.63) | 0.58 (0.54, 0.62) |
| | Counterfactual | 0.40 (0.36, 0.44) | 0.47 (0.42, 0.51) | 0.46 (0.41, 0.50) |
| | Random | 0.42 (0.38, 0.46) | 0.46 (0.42, 0.51) | 0.49 (0.45, 0.53) |
| significantly_lower_center_of_mass | Cofactual | 0.29 (0.25, 0.33) | 0.25 (0.21, 0.29) | 0.19 (0.16, 0.22) |
| | Counterfactual | **0.22 (0.18, 0.26)** | 0.15 (0.12, 0.18) | 0.12 (0.09, 0.14) |
| | Random | 0.22 (0.19, 0.26) | 0.18 (0.15, 0.21) | 0.11 (0.08, 0.13) |
| significantly_lower_note_density | Cofactual | 0.62 (0.57, 0.66) | 0.58 (0.54, 0.62) | 0.14 (0.11, 0.17) |
| | Counterfactual | 0.29 (0.25, 0.33) | 0.29 (0.25, 0.33) | 0.02 (0.00, 0.03) |
| | Random | 0.36 (0.32, 0.40) | 0.37 (0.32, 0.41) | 0.06 (0.04, 0.08) |
| significantly_lower_rhythmic_density | Cofactual | 0.58 (0.54, 0.62) | 0.51 (0.47, 0.55) | 0.23 (0.19, 0.27) |
| | Counterfactual | **0.17 (0.14, 0.20)** | 0.11 (0.09, 0.14) | 0.02 (0.00, 0.03) |
| | Random | 0.20 (0.16, 0.23) | 0.20 (0.16, 0.23) | 0.06 (0.04, 0.08) |
| soft | Cofactual | 0.58 (0.54, 0.62) | 0.55 (0.51, 0.59) | 0.32 (0.28, 0.36) |
| | Counterfactual | **0.26 (0.22, 0.30)** | 0.18 (0.15, 0.21) | 0.04 (0.03, 0.06) |
| | Random | 0.28 (0.24, 0.32) | 0.21 (0.18, 0.25) | 0.08 (0.06, 0.11) |
| some_verticalities | Cofactual | 0.92 (0.90, 0.95) | 0.91 (0.89, 0.94) | 0.74 (0.71, 0.78) |
| | Counterfactual | **0.81 (0.78, 0.84)** | 0.74 (0.70, 0.77) | 0.64 (0.60, 0.68) |
| | Random | 0.90 (0.87, 0.92) | 0.87 (0.84, 0.90) | 0.76 (0.72, 0.80) |
| within_single_key | Cofactual | 0.62 (0.58, 0.66) | 0.58 (0.54, 0.62) | 0.63 (0.59, 0.67) |
| | Counterfactual | 0.53 (0.48, 0.57) | 0.42 (0.37, 0.46) | 0.54 (0.50, 0.58) |
| | Random | 0.59 (0.55, 0.63) | 0.53 (0.49, 0.57) | 0.60 (0.55, 0.64) |

Table 2: Efficacy comparison across all features.

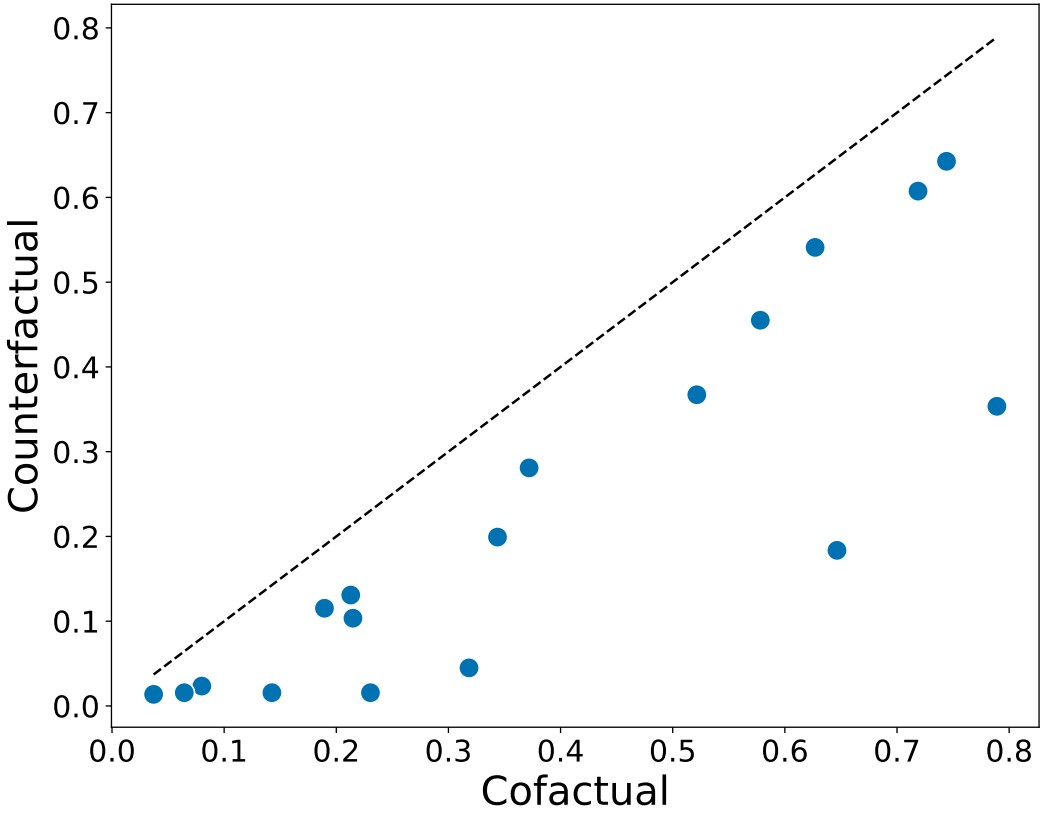

Figure 6: Varying difficulties of features, evaluated with the unconditional model.

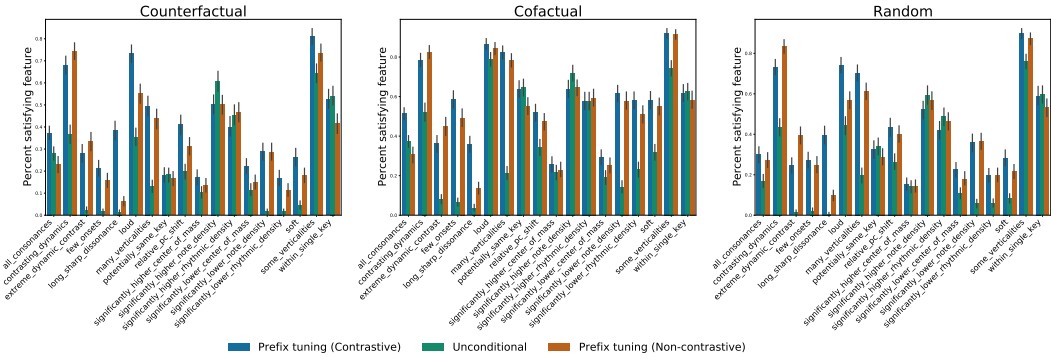

Figure 7: Bar plots.

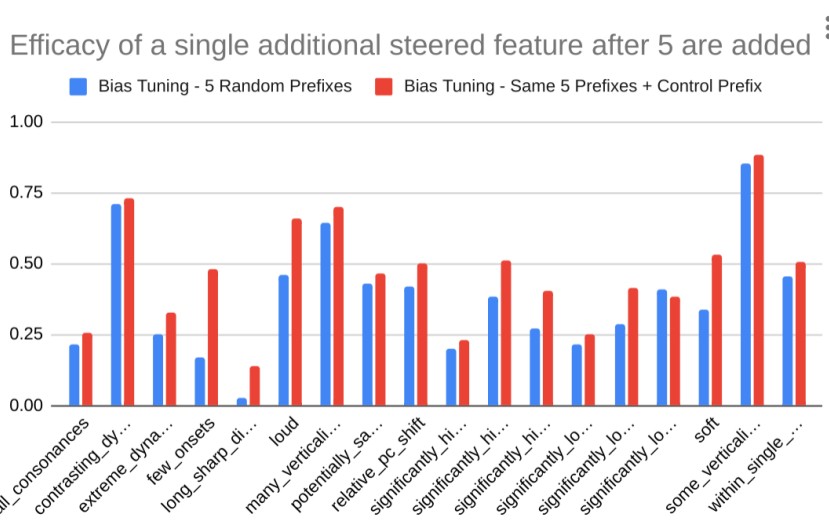

Figure 8: Additional gain after 5 features are already included occur along almost every feature, suggesting that the bias-tuned transformer really can be used compositionally.

