# OpenReview forum: "Composing Features: Compositional Model Augmentation for Steerability of Music Transformers"
_ICLR.cc/2022/Conference — ICLR 2022 Submitted_

### Official Review · Reviewer_LG3E · 2021-10-29

**Correctness:** 4
**Technical Novelty And Significance:** 2
**Empirical Novelty And Significance:** 3
**Recommendation:** 6
**Confidence:** 3

**Main Review:**

My overall sense of this paper is positive, though I am not deeply familiar with the related work on which it builds.
The basic ideas are explained well enough with reasonable motivation and technical justification.
The results seem good, although I'm not sure that I fully understand what is being presented.
In general, the exposition of the methods is clear, but the evaluations (tables and figures) seem unnecessarily terse and difficult to parse.
Putting a bit more explanatory text into the captions and surrounding text would significantly improve this paper. Specifically:

- Figure 4 is described in a one-sentence subsection (5.1), which does not at all explain what is being measured in the figure or how to interpret the plots.  The terms "cofactual" and "counterfactual" are not even introduced until 5.2, but are used in the caption of figure 4 without explanation.  If I understand the plot correctly, it demonstrates that contrastive training eventually assigns high conditional likelihood to "good" continuations (blue) and low likelihood to "bad" ones (green), and therefore is able to discriminate between them?

- The term "efficacy" is used without an explicit definition, and forms the basis of most of the quantitative evaluations (Table 1, Figure 5).  It was only after looking at the page of example outputs that it is clearly defined as the number of rejected samples prior to generating a positive example.  Please define your terms and metrics!

- Table 1 lists what appears to be mean efficacy scores, as well as some kind of bounds in parentheses?  Are these quantiles?

- Figure 5 seems to reuse the same color-map in multiple ways that are inconsistent between plots, making it needlessly difficult to interpret.  (E.g., "unconditional" is blue in the left and green in the middle/right.)


Finally, this is a very minor point, but the neon-green checkmark (e.g., in equation 5) is nearly impossible to read against a white background.
Please do your readers a favor and stick to high-contrast colors.



**Summary Of The Paper:**

This paper describes a method for adapting the parameters of a symbolic music generation model such that generated continuations of a prefix are more likely to satisfy a set of user-specified constraints.
The method builds on prior work by supporting conjunctions (compositions) of constraints, which are treated by embedding constraint specifications as vectors which are provided as input to a bias tuning layer; multiple constraints are handled by averaging their predicted model outputs together.
(A prefix-tuning method is also investigated, but consistently outperformed by the bias-tuning method.)
The model is fine-tuned with a contrastive loss that attempts to maximize the likelihood of continuations that satisfy the supplied constraints while minimizing the likelihood of continuations that violate constraints.
The proposed method is evaluated by its "efficacy", that is, efficiency of generating valid continuations relative to the unconditional model.
A human subject listening test demonstrates that the proposed method is more likely to generate "more musical" continuations than the unconditional model.



**Summary Of The Review:**

It seems like a nice contribution.  It's confusing in parts, but the issues should be easily fixable.

---

> ### Author Response · Authors · 2021-11-23
> **Incorporating Feedback**
>
> Thank you for all of your point addressing clarity, including using terminology before it was introduced and poor explanations.  We have and will continue to improve on these in the paper.
> With regards to the tables, the bounds refer to 95% confidence intervals; We have clarified this in the updated version of the paper.

---

### Official Review · Reviewer_ezPb · 2021-11-02

**Correctness:** 3
**Technical Novelty And Significance:** 2
**Empirical Novelty And Significance:** 2
**Recommendation:** 3
**Confidence:** 4

**Main Review:**

This paper introduces a novel objective to finetune a model on several features (the contrastive loss of Eq. 5) and compares two existing methods (prefix tuning and bias-tuning) to adapt such a pretrained model so that it generates sequences featuring some user-defined musical features. In particular, these two approaches are compared on two different regimes: a single-feature regime where only one musical feature can be selected, and a compositional setting where multiple musical features can be selected at the same time.
If the introduction of the objective of Eq. 6 looks appealing and the two compared methods extracted from recent works, the musical results are not convincing.
The proposed task seems interesting from the machine-learning point of view but looks quite artificial from a musical point of view and it's hard to believe that the musical features of Sect. A may be relevant for composers.

The bibliography seems a bit shallow regarding the relevant literature on generative models for music with (and without) control.
Overall, the paper is well written even if the clarity could be improved at some places (notations in Eq 2 and 3 are not particularly useful, in Fig. 2, there must be a \phi_i as an input in order to compute \ell_x, notations \ell_true and \ell_false are more confusing than helpful as they elude the dependence on \theta_i and \theta_j, a small table showing the difference during training between the single-feature and compositional regimes could help).

Question:
- Influence of alpha on the efficacy of rejection sampling?

**Summary Of The Paper:**

The authors propose to investigate two methods (prefix tuning and bias-tuning) to steer a pretrained music transformer using a set of manually-chosen musical features. The contribution consists in adding custom contrastive loss to the maximum-likelihood objective when finetuning, so that features better capture their intended meaning and offer better control.

**Summary Of The Review:**

There are interesting aspects in this paper (especially the comparison between bias-tuning and prefix-tuning in the compositional setting) but the overall contribution is not very substantial (only Eq. 5) and its effects not extensively discussed. The musical examples are not well presented (impossible to know which are the selected features, only one example per continuation for each model, not obvious to distinguish the generations from the prefixes, extracts are too short) and fail to convince.
The method could be of interest for a musically-focused conference, but it seems that the technical contribution is not strong enough for ICLR 2022.

---

> ### Author Response · Authors · 2021-11-23
> **Incorporating Feedback**
>
> Thank you for your comments. We have significantly added to our related works documentation.
> We have also updated our demo page, where we’ve included longer steered examples (annotated with the features selected, with prime and continuation presented separately) and compositions semi-automatically generated using our feature tuning in a human-directed way. These help illustrate two points:
> How a user/composer might use our method for longer form composition. A composer can control both the short-chunk features, and the long-form structure, by chaining together chunks steered in different directions (using different combinations of features), while maintaining long-term coherence (by leveraging transformers' full self-attention receptive field).
> The longer generations help bring into higher relief the qualitative differences between our method and unconditional generation. Namely, while maintaining musicality, our method is able to satisfy the user-specified musical features; this is unattainable with the unconditional model for pieces of this length.

---

### Official Review · Reviewer_qjek · 2021-11-02

**Correctness:** 2
**Technical Novelty And Significance:** 4
**Empirical Novelty And Significance:** 3
**Recommendation:** 3
**Confidence:** 4

**Main Review:**

I like the CSP setting proposed in Section 2. Much of the related work on steering focuses on soft, differentiable constraints, for example, generating samples that are likely under a classifier model (Dathathri et al., 2020). This is an interesting alternative perspective.

The contrastive loss (Section 3.1) and additive model for feature composition (Section 3.2) are clearly described and interesting. However, the paper is missing a discussion of related work on contrastive losses. How does the contrastive loss proposed here compare to other contrastive losses? I am not very familiar with the literature on contrastive losses, and this lack of discussion leaves me unclear about the extent of the novelty/contribution to the contrastive modeling literature.

I am a little confused by Figure 3: it appears from this figure that prefix-tuning and bias-tuning are used together, but later in the paper it seems that these two techniques are analyzed separately?

I found the discussion of the empirical evaluation (Sections 4, 5, and 6) very confusing. Below are some specific questions and concerns.

(1) Section 4.1 discusses fine-tuning the transformer to the MAESTRO dataset. Does this imply that the original transformer is trained on a different dataset? Which one?

(2) Section 5.2 introduces a lot of terminology with in-line definitions that are difficult to follow. I found it difficult to understand what is meant by "cofactual primes" and "counterfactual primes." What is the distinction made between the "maestro validation dataset" and the "actual validation dataset"? Later, terms "cofactual"/"counterfactual" tuning (Figure 4) and "cofactual steering" (Section 6) are introduced without definition and further confuse me.

(3) Also in Section 5.2: does "efficacy" mean the probability that a sample satisfies the desired constraints?

(4) In Figure 4, what is the y-axis of the contrastive (right-hand) plot? It seems to carry the same "MLE Loss" as the left-hand plot. I'm very unclear what is being shown here (this could be related to my confusion about the meanings of "cofactual" and "counterfactual" tuning).

(5) How are the error bars (confidence intervals?) in Table 1 and Figure 5 computed? How many samples are used to calculate efficacy?

(6) How should I interpret the results of the listening test in Section 6. The motivation for this work stated in the introduction is that rejection sampling is slow, and we want to fine-tune to get a better proposal distribution that allows us to sample more efficiently. But the listening test finds that fine-tuning produces higher quality samples than rejection sampling. This makes me uncomfortable with the framing in the introduction.

Overall, regarding the experiments. There does appear to be good empirical evidence here to support the advantage of the proposed contrastive loss over MLE. But there does not seem to be an empirical analysis of the additive approximation proposed in Section 3. Nor is there a clear analysis of prefix-tuning versus bias-tuning.

Minor comment: the references section cites many arxiv pre-prints, rather than conference versions of the papers. In many cases these papers have been published for several years, and citing the pre-print versions seems a little sloppy.

**Summary Of The Paper:**

This work considers how to control sampling from transformer-based autoregressive generative models of music. In particular, it considers constraint satisfaction problems (CSP) given a collection of binary constraints. The technical contributions of this paper are (1) a method for fine-tuning (bias-tuning) a model using a contrastive loss for adaptation to a given CSP (Section 3.1) and (2) an additive approximation to bias-tuning for satisfying combinatorial constraints (Section 3.2).

**Summary Of The Review:**

I found this paper thought-provoking and creative, but I found the empirical analysis confusing and incomplete. I like the CSP setting outlined in Section 2, and the methods introduced in Section 3 are interesting. However, I found the writing in Sections 4 and 5 confusing to the point that it was difficult for me to understand the empirical results.

---

> ### Author Response · Authors · 2021-11-23
> **Incorporating Feedback**
>
> Literature:
>
> We agree that our submission needs to be better contextualized within the contrastive learning literature, thank you for pointing this out. We added the following to our related work section, which we hope clarifies the similarities and differences between our work and related contrastive learning work.
>
> Contrastive learning is used in representation learning to train a network which maps "similar" (positive) inputs to nearby representations and "dissimilar" (negative) inputs far away from the positive inputs. See Le-Khak et al. (2020) for a theoretical framework and overview. In generative modeling, contrastive divergence (Hinton, 2002) was proposed to train Restricted Boltzmann Machines (Smolensky, 1986), image-to-image translation models (Baek et al., 2021; Park et al., 2020; Liu et al., 2021), and conditional (Kang et al., 2020) and unconditional (Jolicoeur-Martineau, 2018) generative adverarial networks (Goodfellow et al., 2014). Our contrastive formulation differs from previous work in two ways. First, rather than selecting positive and negative examples related to the conditioning signal (musical features in our case) and using the contrastive loss to predict which example "agrees" with the signal, we select positive and negative _conditioning signals_ (i.e. different musical features) and use the contrastive loss to predict which conditioning signal explains the prime–continuation pair best. Second, we also treat the absence of a conditioning signal (i.e. the original generative model) as a negative conditioning signal, meaning that we want the model conditioned on the "correct" musical features to explain the prime–continuation pair better than the unconditional model.
>
> References:
>
> - Kyungjune Baek, Yunjey Choi, Youngjung Uh, Jaejun Yoo, and Hyunjung Shim.  Rethinking the truly unsupervised image-to-image translation. In Proceedings of the IEEE/CVF International Conference on Computer Vision, pp. 14154–14163, 2021.
> - Ian Goodfellow, Jean Pouget-Abadie, Mehdi Mirza, Bing Xu, David Warde-Farley, Sherjil Ozair, Aaron Courville, and Yoshua Bengio. Generative adversarial nets. Advances in neural information processing systems, 27, 2014.
> - Geoffrey E Hinton. Training products of experts by minimizing contrastive divergence. Neural computation, 14(8):1771–1800, 2002.
> - Alexia Jolicoeur-Martineau.   The relativistic discriminator:  a key element missing from standardgan. arXiv preprint arXiv:1807.00734, 2018.
> - Minguk Kang and Jaesik Park.  Contragan:  Contrastive learning for conditional image generation. arXiv preprint arXiv:2006.12681, 2020.
> - Phuc  H  Le-Khac,  Graham  Healy,  and  Alan  F  Smeaton. Contrastive  representation  learning:  A framework and review. IEEE Access, 2020.
> - ​​Rui Liu, Yixiao Ge, Ching Lam Choi, Xiaogang Wang, and Hongsheng Li. Divco:  Diverse con-ditional image synthesis via contrastive generative adversarial network. In Proceedings of the IEEE/CVF Conference on Computer Vision and Pattern Recognition, pp. 16377–16386, 2021.
> - Taesung Park, Alexei A Efros, Richard Zhang, and Jun-Yan Zhu. Contrastive learning for unpaired image-to-image translation. In European Conference on Computer Vision, pp. 319–345. Springer, 2020.
> - ​​P. Smolensky.Information Processing in Dynamical Systems: Foundations of Harmony Theory, pp.194–281. MIT Press, Cambridge, MA, USA, 1986. ISBN 026268053X
>
> Review question: Section 4.1 discusses fine-tuning the transformer to the MAESTRO dataset. Does this imply that the original transformer is trained on a different dataset? Which one?
> Response: It is worth clarifying the model and dataset we used for our work. Although both were provided by the Magenta team, the [model checkpoint was trained on transcribed YouTube piano performances](https://magenta.tensorflow.org/piano-transformer), but this data is not publicly available; on the other hand, they do provide the [Maestro dataset](https://magenta.tensorflow.org/datasets/maestro) (classical virtuosic performances, recorded from real playing as opposed to transcriptions, at a finer grain time resolution), but as far as we know this dataset was not used during the training of the provided Music Transformer model. Indeed, our experiments (see figure 5) show that the Maestro dataset is “out of distribution” with respect to the provided Music Transformer model.

---

> ### Author Response · Authors · 2021-11-23
> **Answering questions**
>
> Reviewer question: I am a little confused by Figure 3: it appears from this figure that prefix-tuning and bias-tuning are used together, but later in the paper it seems that these two techniques are analyzed separately?
> Response: You are correct - the two techniques are used separately. Given that both techniques are applied on the same Music Transformer model, Figure 3 is meant to illustrate both concurrently. We agree this may be confusing, so we will clarify this in the caption of Figure 3.
> Reviewer question: I am a little confused by Figure 3: it appears from this figure that prefix-tuning and bias-tuning are used together, but later in the paper it seems that these two techniques are analyzed separately?
> Response: You are correct: the two techniques are used separately. Given that both techniques are applied on the same Music Transformer model, Figure 3 is meant to illustrate both concurrently. We agree this may be confusing, so we will clarify this in the caption of Figure 3.
>
> Reviewer question:
> Section 4.1 discusses fine-tuning the transformer to the MAESTRO dataset. Does this imply that the original transformer is trained on a different dataset? Which one?
> Response:
> It is worth clarifying the model and dataset we used for our work. Although both were provided by the Magenta team, the [model checkpoint was trained on transcribed YouTube piano performances](https://magenta.tensorflow.org/piano-transformer), but this data is not publicly available; on the other hand, they do provide the [Maestro dataset](https://magenta.tensorflow.org/datasets/maestro) (classical virtuosic performances, recorded from real playing as opposed to transcriptions, at a finer grain time resolution), but as far as we know this dataset was not used during the training of the provided Music Transformer model. Indeed, our experiments (see figure 5) show that the Maestro dataset is “out of distribution” with respect to the provided Music Transformer model.
>
> Reviewer question: Section 5.2 introduces a lot of terminology with in-line definitions that are difficult to follow. I found it difficult to understand what is meant by "cofactual primes" and "counterfactual primes." What is the distinction made between the "maestro validation dataset" and the "actual validation dataset"?
> Response:
> Thank you for pointing this out, we meant “Maestro validation dataset” where we had written “actual validation dataset”. We have corrected this in the paper.
> Later, terms "cofactual"/"counterfactual" tuning (Figure 4) and "cofactual steering" (Section 6) are introduced without definition and further confuse me.
> Thank you for pointing this out, we have added a reference to section 5.2 (where these terms are introduced) to the caption of Figure 4.
>
>
> Review question: Also in Section 5.2: does "efficacy" mean the probability that a sample satisfies the desired constraints?
> Response: Yes, we can clarify that.
>
> Reviewer comment: "In Figure 4, what is the y-axis of the contrastive (right-hand) plot? It seems to carry the same "MLE Loss" as the left-hand plot. I'm very unclear what is being shown here (this could be related to my confusion about the meanings of "cofactual" and "counterfactual" tuning). "
> This plot shows the NLL likelihood of the actual continuations using the Maestro dataset, and given different types of steering - cofactual steering (steering towards a feature that the Maestro continuation possesses), counterfactual steering (steering towards a feature that the Maestro continuation doesn’t possess), and no steering at all.
>
> Reviewer Question: How are the error bars (confidence intervals?) in Table 1 and Figure 5 computed? How many samples are used to calculate efficacy?
> Response: We use 95% confidence intervals, and 5 iterations per sample prime with (the same) 256 sample primes in each setting.
>
> Reviewer question:
> "How should I interpret the results of the listening test in Section 6. The motivation for this work stated in the introduction is that rejection sampling is slow, and we want to fine-tune to get a better proposal distribution that allows us to sample more efficiently. But the listening test finds that fine-tuning produces higher quality samples than rejection sampling. This makes me uncomfortable with the framing in the introduction."
> Response:
>  While our main goal was to sample more efficiently, we found that in cases when the steering was most necessary, the few examples which the unconditional model produced were of significantly lower quality.
> Our approach may produce more consistent generation due to the way it is contrastively trained, but that is not in itself the only contribution of our paper.  Rather, we attempt to produce more plausible outputs which satisfy the given constraints per generation attempt.
> Also, please refer to our new demo page, where we’ve included longer steered examples and compositions semi-automatically generated using our feature tuning in a human-directed way.

---

> ### Author Response · Authors · 2021-11-23
> **Additional question addressed**
>
>
> Reviewer Comment:
> "Overall, regarding the experiments. There does appear to be good empirical evidence here to support the advantage of the proposed contrastive loss over MLE. But there does not seem to be an empirical analysis of the additive approximation proposed in Section 3.
> Nor is there a clear analysis of prefix-tuning versus bias-tuning."
> Response:
> In the prefix-tuning setting we compared concatenation versus averaging and found that averaging gave better performance.
> In the bias-tuning case, because we are modifying the bias terms of the network (which have a fixed shape), we only perform averaging, since concatenation is no longer a possibility.
> In Figure 5 we compare prefix-tuning (with concatenation and averaging) with bias-tuning, and observe much better performance with bias-tuning.

---

### Official Review · Reviewer_ZYeZ · 2021-11-03

**Correctness:** 2
**Technical Novelty And Significance:** 3
**Empirical Novelty And Significance:** 2
**Recommendation:** 3
**Confidence:** 3

**Main Review:**

This paper tackles an important topic of music generation in the context of Human-AI co-creation. Steering the generation results to the direction intended by the user has been emphasized as a lesson of recent AI song contest events. The authors address the issue focusing on continued generation given a prime. The proposed contrastive learning is novel and the superiority is explained through the experiment. The demo examples support the effectiveness in musicality. However, the presentation and results are not clear to me in term of validating steerability of the model.

- While the demo examples contrasts the superiority of the proposed approach well, I have a bit doubt about the results. The music transformer model is known to tackle the long-term dependency issues in music generation. Using the transformer architecture and relative positional encoding, the model is known to learn wider music contexts and generate consistent note patterns given a prime. However, the majority of unconditional generation examples from Music Transformer diverge in the continuation. On the other hand, the bias-tuning model sounds very consistent between the primes and the continuations. This is somewhat confusing because I was expecting that Music Transform is good at consistent generation whereas bias-tuning may steer the generation to different textures which could not be consistent with the prime, for example, when block chord is chosen as a musical feature for the continuation and the prime does not have block chords. I think the authors should put "consistent generation" in the thread of discussion and clarify how steering is different from it.

- The listening test compares unconditional generation to conditional generation with "randomly chosen three-feature sets".  This experiment does not explicitly validate the controllability of the model. That is, it does not test steering the generation by the user. I think, if the purpose was rating the preference,  general music listeners could have done it instead of musicians. The user test should be designed such that the musicians choose one of the musical features that they want to control and then rate if the conditional generation is satisfactory or aligned to their expectation.

- "Non-contrastive" loss is compared to the proposed contrastive loss in Figure 4. But, it seems that the non-contrastive loss is not clearly defined in the paper. Is it the loss that does not include the negative case (this is my guessing)?

- 12 absolute musical features and 8 relative musical features are defined in Appendix A. I see that they are defined based on music elements such as dynamics, tonality, polyphony, note density and so on. However, they are mostly low-level attributes and are not a complete list of musical elements. Furthermore, more high-level features are possible, for example, syncopation, chromaticism, more jazz chords (e.g, more 7ths chords) but they are not addressed. Authors should clarify the basis of selecting the music features and the scope of possible musical steering with the features.

- This motivation and necessity of the research was well addressed in the introduction. However, the author missed the following paper which testifies the labor-intensive rejection sampling in the real-world music generation scenario.

** Human-AI Co-creation in Songwriting, Cheng-Zhi Anna Huang, Hendrik Vincent Koops, Ed Newton-Rex, Monica Dinculescu, Carrie Cai, ISMIR 2020.

There will be more papers that reported similar issues in the context of computational creativity.

< Minor parts>
- page 2: "... for instance, 6, the requested... " --> "6" seems to be an errata.
- page 3: the green symbols (the check marks) in section 3.1 are not visible when it is printed in black and white.
- page 4: "prepreding" --> This looks like a typo. "prepending" might be correct


**Summary Of The Paper:**

This paper presents a model tuning strategy for steering conditional music generation given the pre-trained Music Transformer model. The authors pose the steering as musical feature matching between two sequences from prime and continuation, respectively, and tune the parameters using two methods, prefix tuning and bias-tuning. They show that the latter is more effective for steering conditional music generation. They also present a differentiable proxy of music feature matching considering the positive, negative, and non-adaptive cases. The contrastive loss outperforms the non-contrastive loss and bias-tuning is more effective than prefix tuning in the experiment.

**Summary Of The Review:**

This paper tackled an important issue of music generation. They proved the proposed contrastive loss by compared it to the non-contrastive loss. However, while the aim of the research is providing controllability or steerability of the model by musical features, the listening test seems to validate preference of the continuation. In other words, the listening test does not reflect the control by the musician subjects. The authors should clarify this during the discussion phase.

---

> ### Author Response · Authors · 2021-11-23
> **Incorporating feedback**
>
> We thank the reviewer for their thoughtful feedback.
>
> In several examples, a human used this generation enhancer to generate several compositions. The human used their own discretion in choosing prefixes, starting material, and which of the many generated elements per batch to continue with, but did not otherwise alter the music.
>
> Reviewer1P1. _Disconnect between the observed long-term consistency of Music Transformer and the unconditional continuations on one hand._
> Reviewer1P2. _On the other hand, the bias-tuning model sounds very consistent between the primes and the continuations. Authors should put "consistent generation" in the thread of discussion and clarify how steering is different from it._
>
> Response: We agree with the reviewer that this seems counterintuitive. There are a few factors at play here.
> It is worth clarifying the model and dataset we used for our work. Although both were provided by the Magenta team, the [model checkpoint was trained on transcribed YouTube piano performances](https://magenta.tensorflow.org/piano-transformer), but this data is not publicly available; on the other hand, they do provide the [Maestro dataset](https://magenta.tensorflow.org/datasets/maestro) (classical virtuosic performances, recorded from real playing as opposed to transcriptions, at a finer grain time resolution), but as far as we know this dataset was not used during the training of the provided Music Transformer model. Indeed, our experiments (see figure 5) show that the Maestro dataset is “out of distribution” with respect to the provided Music Transformer model.
> * When sampling continuations conditioned on Maestro priming sequences, we are doing out of distribution generation, and it is not entirely unexpected that generation could fail in that setting.
> * In addition to sampling continuations from out-of-distribution priming sequences, we are also steering the model through rejection sampling towards satisfying musical features, which further skews the distribution.
> * Furthermore, as we demonstrate in the demo, the instances of unconditional generation which satisfy unlikely constraints tend to sound more inconsistent than ones which are unsteered.
> We thank the reviewer for raising these points, and we will clarify these points in the paper.
>
> Reviewer1P3: _Listening test does not explicitly validate the controllability of the model._
>
> Response: There are two main components to validate:
> * Steering effectiveness, which we can validate algorithmically by measuring the acceptance rate for a given musical feature.
> * Musical quality, which is subjective and therefore hard to measure numerically.
>
> We verified point #1 in Figures X, Y, Z in the paper, where we can assert that our method is more effective at steering (in statistically significant manner).
> Point #2 is difficult to measure numerically, so we used the listening test for this purpose. We specifically chose random features so as to not bias the measurements of musical quality.
>
> Further, we added long-form compositions (see demo page.)

---

### Author Response · Authors · 2021-11-23
**Thanks for your insights; addressing common concerns**

# Overall response

We would like to thank all the reviewers for the careful reading of the paper and their thoughtful comments and suggestions. While we are responding to each reviewer separately, we wanted to highlight our response to a few common concerns raised by multiple reviewers. When making changes to the paper based on reviewer comments, we have used blue text to clarify what has changed since the original submission.



* Many reviewers found it difficult to understand some of the qualitative results we provide. To address this, we have:
    * clarified notation in places where reviewers raised issues
    * clarified the motivation for the original listening tests included in the initial submission (testing for musical quality, not efficacy; efficacy is evaluated by the quantitative experiments included in the initial submission)
    * added a more detailed description to the demo page
    * added long-form composition examples to the demo page to highlight the usefulness of our method in crafting longer-form songs, and further demonstrate the difficulty in simply using the pre-existing transformer model
* Several reviewers expressed concerns with the clarity of empirical evaluations. To address this, we have

    * modified the text as per suggestions by the reviewers
    * clarified why the results produced by the pre-trained Music Transformer model are of not very high quality
* Some concerns were raised with the connections drawn to previous works. To address this, we have drawn stronger connection to prior work in:
    * contrastive learning to motivate our contrastive approach
    * (controllable) music generation to motivate the need for steering pretrained music language models in a compositional way

We again thank the reviewers, and welcome any feedback.

---

### Decision · Program_Chairs · 2022-01-20

**Decision:**

Reject

**Comment:**

This paper proposed a compositional approach to (conditionally) steer pre-trained music transformers to the direction intended by the user.  Overall the scores are mostly negative. The reviewers pointed out some interesting aspects of the paper (e.g., using hard binary constraints as opposed to the soft ones, the contrastive approach). However, one common issue shared by all the reviewers is the clarity of the presentation, which led to many reviewers being confused about various aspects of the paper especially the empirical evaluation. The authors did provide a detailed response to address some of the concerns, but to fully address all the points I anticipate it would require quite substantial change to the paper. A couple reviewers also raised the concerns regarding the limited contribution of the paper. Finally, there appears to be some disagreement between the authors and reviewers regarding how to interpret the listening test results. I hope the authors can take the comments into consideration to further improve this paper for the next submission.